# The Potential Material Flow of WEEE in a Data-Constrained Environment—The Case of Jordan

Laila A. Al-Khatib [1] and Feras Y. Fraige [2,3,*]

1 Environmental Engineering Department, Faculty of Engineering, Al-Hussein Bin Talal University, P.O. Box 25, Ma'an 71111, Jordan; laila@ahu.edu.jo
2 Mining & Minerals Engineering Department, Faculty of Engineering, Al-Hussein Bin Talal University, P.O. Box 25, Ma'an 71111, Jordan
3 Mechanical Engineering Department, Faculty of Engineering, Al-Hussein Bin Talal University, P.O. Box 25, Ma'an 71111, Jordan
* Correspondence: ferasfraige@gmail.com; Tel.: +962-788564979 or +962-32179000 (ext. 8505); Fax: +962-32179050

**Abstract:** The rising concerns about electric and electronic equipment waste (WEEE) come from the rapid increase in demand for appliances and the decreasing lifetimes of equipment. Setting a sustainable WEEE management system that exploits this secondary resource is paramount to maximize resource efficiency, mitigate its environmental impact, and stimulate the circular economy. This paper aims, for the first time, to quantify the material flow expected from recycling the generated WEEE, propose the number of plants required to recycle this secondary resource, and outline the expected economic and environmental benefits that could be achieved from recycling operations. The findings of material flow calculations show that the amount of steel, copper, and aluminum is predominant in the WEEE composition. Also, the expected metal content in WEEE in 2022 is approximately 26 kt, 3.3 kt, and 2.5 kt, respectively. These are expected to substantially increase to approximately 109 kt, 11.9 kt, and 9 kt for the three metals in 2050, respectively. Other valuable metals are doubling their quantities between 2022 and 2050 to reach approximately 1133 kg silver and 475 kg gold. Approximately, four treatment plants are required to recover these materials in 2030 with relative installation costs of USD 100 million. The forecasted financial revenues of recovering materials included in WEEE and indicators for environmental impact based on life cycle assessment (LCA) are calculated. The results of this study can serve as a preliminary reference for future usage in guiding effective planning for WEEE recycling and sustainable management in the country.

**Keywords:** WEEE; recycling; sustainability; population balance model; material flow; environment



## 1. Introduction

The production and consumption of electric and electronic equipment (EEE) are growing fast at the global level [1–3]. Further, factors such as higher economic growth, urbanization, industry 4.0, and technological advancement lead to rapid product obsolescence [1]. As a result, the waste of these devices is hitting a global record of 53.6 Mt in 2019, an increase of 21% compared with 2014 [2]. And it is not away from doubling 2014 figures by 2030 [3].

EEE contains a wide spectrum of products, designs, and brands that includes, according to EU-6 classification, temperature exchange equipment (like refrigerators "Rs" and air conditioners "ACs"), large equipment (such as washing machines "Ws" and dryers), small equipment (like food processors and vacuum machines), screens and monitors (like televisions "TVs" and laptops), small IT equipment (like external drives, and keyboards), and lamps (like straight tube and compact fluorescent) [2]. These products have assorted sizes, weights, compositions, and values even for the same type [4]. Eventually, after a certain time, they will reach their end of life (EoL). These obsolete appliances are called

waste electric and electronic equipment (WEEE). It is, indeed, a unique waste due to its diversity and complexity in terms of its constituents. WEEE contains valuable metals and hazardous materials [5]. Base metals, including iron, steel, aluminum, copper, zinc, and their alloys, form the majority by weight. Precious metals, such as gold, silver, and palladium, are used in low proportions especially in printed circuit boards (PCBs), while plastics represent 15–35 percent of WEEE [6–8]. WEEE also contains small amounts of critical metals, including Indium, Cobalt, Gallium, Tungsten, and various rare earth elements (REEs) [9–11]. REEs are essential for attaining the appropriate level of technological advancement in the manufacturing of modern electronic equipment [10–13]. On the other hand, the presence of hazardous substances with its profound potential for toxicity (such as lead, mercury, cadmium, brominated flame retardants (BFRs)) and environmental impact (such as chlorofluorocarbons (CFCs) and other coolants) requires handling WEEE effectively and separately from other solid waste streams [14,15].

Material composition for each EEE type is a principal factor in analyzing material flow. Different studies have investigated the material composition of these appliances including large and small household appliances, IT products, screens, and monitors [14,16–19]. The conclusions of these studies indicated that there is great variability, even for the same appliance, following different designs, brands, locations, and manufacturing dates among others [4]. So, to minimize this heterogeneity and consequence uncertainty in data, samples should be large enough to consider the aforementioned factors.

Consequently, it is imperative to set reverse and circular path policies/strategies to ensure these valuables are sustainably recovered [20,21]. The potential of appliance repair, reuse, recycle, remanufacture, redesign, and rethinking of other uses should be explored to maximize materials exploitation. This will certainly extend the life of these appliances and can have many economic, environmental, and societal benefits [5,19]. Firstly, the side effects of mining activities searching for raw materials will be reduced. Also, environmental problems associated with inadequate dumping and recycling can be mitigated. Reuse and recycling of WEEE decreases the amount of waste sent to landfills and combat illegal export to developing and underdeveloped countries (where regulations and laws, if present, are lax). Finally, applying circular strategies creates jobs and reduces imports of materials which is considered a prospective growth for a country [20,21]. Usually, developing a sustainable and efficient WEEE management policy requires reliable estimation of waste generation [22,23]. So, it is vital to forecast the generated WEEE and the potential material flow from its recycling.

There are various methods available in the literature to quantify WEEE generation. For example, input–output analysis, projections analysis, disposal analysis, and factor models [21,24–28]. The input–output analysis evaluates material flow routes from sources to the destination [29]. Combined with average appliance material composition, metal flow contained in WEEE can be analyzed [30,31]. In particular, the population balance model (PBM) is used to predict WEEE generation patterns [21,32–36]. Because PBM originated from mass balance principles, where inflow is EEE shipment volume; outflow is WEEE volume; stock is EEE ownership, the generated WEEE is not prone to over or underestimate. In addition, it presents invaluable information about material flow with time that enables predictions whether it is in past or future periods. However, in some circumstances where the product is in fast growth or decline stages, difficulties in estimating parameters may be faced. Thus, careful procedures should be taken [21,32,33].

WEEE processing employs several methods to handle and recycle the waste in an ecologically responsible and sustainable way. The processing techniques may differ based on the nature of generated waste and local circumstances. They include collection and sorting, dismantling, shredding, mechanical separation, hydrometallurgical processes, pyrometallurgical processes, biological treatment, incineration and energy recovery [5,9–12,17,20,22]. Japan, Germany, South Korea, Taiwan, Switzerland, and Norway are among the worldwide well-advanced countries in the field of WEEE recycling. For example, in 2010, Japan witnessed the largest treatment of WEEE collected during the 1990s and 2000s. Approximately

25.8 million units were recycled out of 27.7 million units discarded in 49 processing facilities [32,37]. On average, and given today's technological advancement, these processing plants can treat approximately half a million appliances annually. Many successful recycling plants for TVs, Rs, Ws, and ACs are in action now in Japan with resource recycling efficiencies surpassing 94% [38]. They are exceeding the legal recycling guidelines outlined in the Japanese Home Appliance Recycling Law (HARL) [39]. For instance, Panasonic Eco Technology Center (PETEC), has superseded legal baseline requirement and obtained higher recycling and resource rates. PETEC uses three terms to describe their system's efficiency: "material efficiency" ($\eta_M$), "thermal efficiency" ($\eta_T$), and "resource efficiency" ($\eta_R$). $\eta_M$ is defined as the percentage that is used for material recycling whether it is reusing whole/parts/components or raw materials for new products, whereas $\eta_T$ is the efficiency resulting from using heat generated from incinerating disposable parts. The latter, $\eta_R$, refers to the overall efficiency including material and thermal efficiencies (i.e., $\eta_R = \eta_M + \eta_T$) [38].

In summary, there are many economic, environmental, and social gains from WEEE recycling. Setting a sustainable WEEE management system that exploits this secondary resource is paramount to maximize resource efficiency, mitigate its environmental impact, and stimulate the circular economy.

*Motivation of This Study*

In the least developed nations, economic development is often prioritized over environmental conservation [40]. Developing countries, like Jordan, often face problems related to the establishment of sustainable waste management systems. These include, but are not limited to, the absence of reliable waste inventory records, inefficient or uncomprehensive regulations controlling waste handling/treatment/disposal, lack of appropriate technological infrastructure, and deficiency of financial support [20,21]. For practical and efficient WEEE management systems, these obstacles should be eliminated, or at least minimized. The quantification of material flow from generated WEEE is essential for possible recycling activities. These activities can be carried out through two routes. Either WEEE can be collected and exported legally to countries with advanced processing capabilities. Alternatively, recycling activities can be performed within the country after establishing proper recycling plants and ensuring that the workforce is adequately trained. Hence, the particular importance of this paper lies in filling these gaps which will be of benefit to Jordan and countries sharing the same status and issues.

Thus, some of these limitations will be tackled and proper recommendations will be outlined. The main objective in this paper is to evaluate the material flow included in WEEE using PBM giving valuable insight into the time series of disposed materials. Additionally, to bridge the gap related to the absence of appropriate WEEE treatment plants in Jordan, the required number of treatment plants and the relative cost of installation will be approximated. Finally, the expected financial revenues, environmental, and social benefits are discussed.

## 2. Materials and Methods

In this paper, PBM is used to assess the material flow of WEEE in Jordan. The methodology of PBM is further elaborated in the subsequent sections.

### 2.1. Data Sources

Historical data about population and EEE are obtained from Jordan's Department of Statistics (JDOS) website [41]. Import, export, and production data are extracted from the harmonized commodity description and coding system of each selected EEE from the JDOS online website and UN Comtrade Databases [42]. This study uses survey results for the average lifetime of EEE as summarized in Table 1 [40]. Household size, household number, and percentage share of appliances in Jordanian households are obtained from the predictions conducted by Fraige et al. [21].

**Table 1.** Average EEE characteristics of baseline scenario (BL) in this study.

| EEE Type | Average Lifetime (Years) | Average Weight (kg) | Average Material Composition | | | | | | | |
|---|---|---|---|---|---|---|---|---|---|---|
| | | | Fe (%) | Cu (%) | Al (%) | Ag (PPM) | Au (PPM) | Pd (PPM) | Plastics (%) | Others (%) |
| CRT TV | 10.5 | 33.2 | 10.3 | 3.7 | 2.6 | 12 | 0.5 | 2 | 22.8 | 60.6 |
| FPD TV | 10.5 | 14.7 | 46.9 | 3.8 | 4.7 | 58.2 | 24.5 | 15.3 | 24.2 | 20.4 |
| R | 11.8 | 69.50 | 61.7 | 3.4 | 2.5 | NR [1] | NR | NR | 27.8 | 4.6 |
| W | 9.3 | 72.90 | 52.1 | 1.9 | 3.1 | 0.19 | 0.06 | NR | 6.8 | 36.1 |
| AC | 8.3 | 28.00 | 54.4 | 15.6 | 9.4 | NR | NR | NR | 15.7 | 4.9 |

[1] NR: not reported.

### 2.2. Scope of This Study

In this study, attention is focused on the material flow of WEEE generated from the most used household appliances. These appliances cover the main categories such as washing machines to represent large home appliances, TVs for monitors and screens, refrigerators, and air conditioners for temperature exchange equipment. The rationale behind selecting this particular combination of equipment is as follows. Firstly, high rates of diffusion of these appliances have been observed in developing countries during economic growth periods [43]. Secondly, they represent the expected great percentage of Jordan's WEEE stream [21,40]. Thirdly, several researchers have chosen these EEE to approximate national WEEE generation rates in different countries [32,33]. However, it is acknowledged that the generation of WEEE should ideally encompass all EEE. And somehow the inclusion of this combination may be considered as a limitation of this study. Nevertheless, when it comes to reality, these EEE have the highest percentage share in society, and they are bulky and require space for storage once reach their EoL, hence, high collection rates are expected. All these reasons encourage the adoption of these EEE to approximate the WEEE in the country.

### 2.3. PBM

PBM is a widely used tool in predicting WEEE generation rates proposed by Tasaki et al. [34,35] and employed more recently by different researchers [21,32,33,36]. It is successfully applied to estimate WEEE in both developed countries such as South Korea [33] and developing countries such as Vietnam [32] and Jordan [21]. In this model, the change in appliance numbers possessed between two successive years ($N_T$ and $N_{T-1}$) is presumed to balance the resultant total shipment ($S_T$) minus the total number discarded ($D_T$) in year $T$. It can be expressed by the following equation [34,35]:

$$N_T - N_{T-1} = S_T - D_T \tag{1}$$

The possession number of each appliance's type at a specific year ($N_T$) can be estimated using the following equation:

$$N_T = n\,H \tag{2}$$

where $n$ is the share percentage of appliance per household, and $H$ is the number of households, obtained from [21]. EEE shipment figures are obtained from past data collected from previous statistics and literature as indicated in data sources. In addition, future shipment figures are systematically predicted by PBM. EEE disposal is correlated with purchased appliances and their average lifetime to disposal incorporating the Weibull probabilistic distribution function as recommended by [29,44]. The disposal ratio ($d_t$) of EEE at a certain age $t$ can be estimated by the following equation [45]:

$$d_t = \frac{\beta}{\alpha}\left(\frac{t}{\alpha}\right)^{\beta-1} e^{-\left(\frac{t}{\alpha}\right)^{\beta}} \tag{3}$$

where $\alpha$ and $\beta$ are the Weibull distribution parameters. The scale parameter ($\alpha$) is the lifetime of each EEE. While the shape parameter ($\beta$) is assumed 3.44 (i.e., the case where mean = median for normal distribution [46]). The discarded number of EEE at specific year $T$ denoted by $D_T$ is estimated from the summation of all discarded EEE from the year that the appliance is put on the market ($T_p$) up to year ($T - T_p$) assuming that $d_t$ is zero when $T = T_p$. This will streamline the calculation process, particularly for future forecasts of EEE put on market, depending on the demand for EEE and PBM. So, $D_T$ can be estimated based on the following equation [45]:

$$D_T = \sum_{T_p < T} S_{T_p} \cdot d_{(T - T_p)} \tag{4}$$

### 2.4. Materials Included in WEEE

Wang conducted comprehensive research gathering representative material compositions for various appliances from distinct locations, encompassing several models, of different sizes and shapes over extended periods [4]. This invaluable database of EEE weight and composition can be considered a representative sample of EEE especially in countries where such data are unavailable. So, it is adopted in this work. A summary of the EEE weight and composition for the baseline scenario (BL) is shown in Table 1. It is worth mentioning that flat panel display television sets (FPD TV) were replacing cathode ray tube television sets (CRT TV) in the 2000s. According to Jordanian trade experts, the sales of CRT TV sets ceased in 2010.

### 2.5. Estimation of WEEE Processing Plants Number

Indeed, Jordan can benefit from the accumulative expertise of countries that have advanced processing capabilities in the field of WEEE recycling. This includes establishing processing infrastructure, knowledge transfer and building capacities in various processing methods (mechanical, pyrometallurgy, hydrometallurgy, etc.), and adopting best practices techniques for recycling WEEE. This can be achieved through well planned WEEE policies and regulations as well as collaboration and cooperation between Jordan and countries with well-established recycling infrastructure. Hence, this article utilizes the Japanese experience of setting up and running WEEE treatment facilities to estimate the required number of processing units to handle the projected WEEE generation in Jordan. On average, and given today's technological advancement, approximately half a million appliances per year can be treated at such a processing plant. The relative cost of each plant was approximately USD 18 million in 2010, and approximately USD 25 million in today's prices (inflation rate of 2.68% per year between 2010 and 2023 is obtained from the US Bureau of Labor Statistics https://www.bls.gov/data/ accessed on 16 November 2023) [32].

### 2.6. Financial Revenue Estimation

Financial revenue is a function of recycling efficiency and the value of recovered materials. According to Japanese processing technology adopted in this work, the legal recycling efficiency ($\eta_L$) mandated in HARL [39] and the optimum values of material recycling efficiency ($\eta_M$) for the different appliances obtained by PETEC [38] are summarized in Table 2. While market prices of the recyclable materials are obtained from the websites of: Management Engineering & Production Services International Ltd. (Sheffield, UK) for ferrous metal [47]; London Metal Exchange (London, UK) for copper, aluminum, and zinc [48]; Kitco Metals Inc. (Montreal, Canada) for gold, silver, and palladium [49]; and Recycling Monster (CA, USA) for plastics [50]. The average price for 2022 is taken as a base year in the calculations as shown in Table 3.

**Table 2.** The efficiencies ($\eta_L$ and $\eta_M$) adopted in this study.

| EEE Type | Legal Recycling Efficiency ($\eta_L$, %) | PETEC Material Recycling Efficiency ($\eta_M$, %) |
|:---:|:---:|:---:|
| TV | 74 | 88 |
| R | 70 | 80 |
| W | 82 | 93 |
| AC | 80 | 95 |

**Table 3.** The average market price (2022 base year, American United States Dollar) of recyclable materials.

| Material | Fe | Cu | Al | Ag | Au | Pd | Zn | In | Plastics | Stainless Steel |
|:---:|:---:|:---:|:---:|:---:|:---:|:---:|:---:|:---:|:---:|:---:|
| Price (USD/kg) | 1.28 | 9 | 2.45 | 770 | 63,500 | 74,500 | 3.18 | 572.1 | 1 | 1.7 |

### 2.7. Environmental Benefits Estimation

Life cycle assessment (LCA) is an internationally recognized and standardized methodology. It primarily comprises four stages: (1) defining the objectives and scope of the life cycle assessment; (2) conducting a life cycle inventory; (3) evaluating the impact of the life cycle; and (4) interpreting the life cycle [51,52]. In general, the purpose of life cycle assessment (LCA) is to evaluate the environmental impact of products and the processes linked to them. LCA includes various application aspects. One prime illustration is that it has been employed to assess a product's life cycle from raw material extraction to product usage and eventual disposal [53]. Other studies have concentrated on a specific product's production stage and during its end-of-life stage, while others have focused on waste management [52,54].

In the scope of this study, environmental benefits are estimated in terms of mitigating the environmental impact of raw material extraction and refining. Using the principles of life cycle assessment (LCA), the environmental impact indicator from the "ReCiPe" method for each element is utilized to estimate scores for the relative severity of a product [55]. These scores are expressed by a point scale (pt) to show the magnitude of the impact. The higher the value in point scale the greater the environmental impact, and thus needs to be given priority in recycling operations. Based on the environmental impact of the primary production of the recyclable materials summarized in Table 4 [55], and given the composition and weight of each EEE, the environmental impact of each device is estimated by adding all impacts from the contained materials together.

**Table 4.** The environmental impact of the primary production of recyclable materials.

| Material | Fe | Cu | Al | Ag | Au | Pd | Plastics |
|:---:|:---:|:---:|:---:|:---:|:---:|:---:|:---:|
| Environmental Impact (pt/kg) | 0.2 | 0.5 | 1.1 | 16.6 | 1540.6 | 9832 | 0.45 |

### 2.8. Sensitivity Analysis and Scenarios

Because the WEEE management and modeling inherently incorporate assumptions and uncertainties, providing precise estimates of WEEE generation and material flow figures are challenging particularly when data is scarce. Many parameters can affect the results obtained including material content in EEE and average lifetime, in addition to other logistic and socioeconomic factors, such as collection rates, behaviors, market dynamics, and income [56]. In the scope of this paper, EEE weight, composition, and lifetime are included in the sensitivity analysis to investigate their impact on WEEE generation, material flow, and potential revenues. Firstly, three different scenarios for EEE weight and composition are considered. Baseline scenario employed data shown in Table 1. While

scenarios 1 and 2 utilized EEE average weight and composition from [32] as scenario 1 (S1) and the EU average weight and composition data from [57] as scenario 2 (S2) as shown in the Appendix A.

On another hand, the amount of generated WEEE depends on the time probability distribution function of the EEE before its disposal. In the scope of this study, the effect of varying Weibull distribution parameters (the scale parameter, $\alpha$; and shape parameter, $\beta$) are investigated by taking ±30% of its original value in the BL scenario for each parameter. The disposal ratio of each EEE is recalculated after changing these parameters. This is fed to the PBM to predict the time series of generated WEEE. These scenarios are summarized in the Appendix A. The relative difference percent (*RD*) between the baseline scenario (*BL*) and the i-th scenario (*Si*) is employed to measure the uncertainty in estimations and it is given by:

$$RD\ (\%) = \frac{Si - BL}{BL} \times 100 \tag{5}$$

## 3. Results

### 3.1. WEEE Generation Rates

The amount of generated WEEE from the studied appliances in Jordan under baseline scenario is predicted up to 2050. The number of obsolete appliances is expected to follow an increasing trend with time as illustrated in Figure 1. In 2022, total appliance disposal is estimated to reach 1.6 million units. By 2044 and 2050, the number of WEEE will have doubled and tripled from their predictions in 2022. The predicted weight of WEEE is expected to undergo an almost 3.7-folds increase, rising from 53 kt in 2022 to 198 kt in 2050.

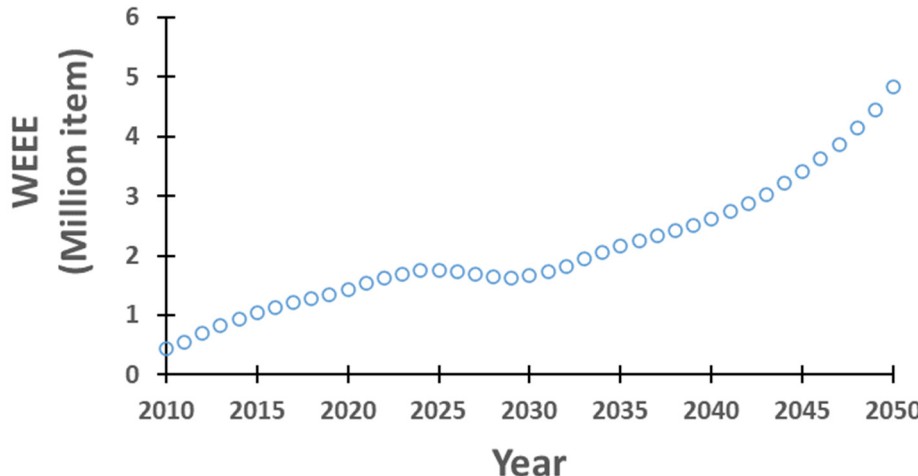

**Figure 1.** Expected rate of WEEE generation by number in Jordan.

### 3.2. WEEE Material Flow

This section provides the findings of the anticipated flow of materials within the WEEE stream under baseline scenario, with a specific emphasis on ferrous metals, copper, aluminum, silver, gold, and plastics. While other materials are found in the examined EEE, the materials listed are the most abundant.

The content of steel in WEEE generated in Jordan during the study period is illustrated in Figure 2. It shows an increasing trend of steel weight enclosed in the generated WEEE with time from 26 kt (49% of WEEE) in 2022 to approximately 109 kt (55% of WEEE) in 2050. This represents a growth of approximately 4.2 folds 2022 figures. It also observed that steel percentage in the discarded WEEE depends on the type of the investigated appliances. For example, steel composition in Ws increases from 22.4% (5.8 kt) in 2022 to more than 35.2% (38.4 kt) in 2050 as a percentage in the waste stream. A slight steel content percentage increase in Rs in the WEEE stream to approximately 33.3% (36.3 kt) is predicted in 2050 from 2022 figures (32.1%, 8.4 kt). On the contrary, the percentage of steel in WEEE originated

from discarded TVs, and ACs decreases with time even though the weight of steel in all discarded appliances is increasing with time. In 2022, the percentages for TVs, and ACs are 18.8% (4.9 kt) and 26.7% (7 kt), respectively. The expected decline in 2050 will be drastically observed in TVs' steel contents with more than half of 2022 figures (8.3%, 9 kt). AC steel content is expected to reach 23.2% (25.3 kt) in 2050.

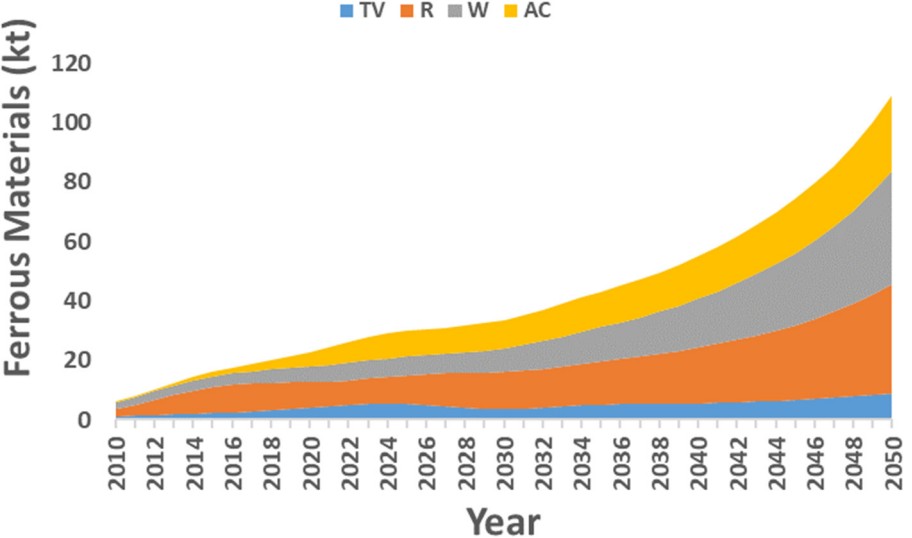

**Figure 2.** Ferrous materials contained in the discarded WEEE in Jordan.

Similarly, it is anticipated that the weight of copper contained in WEEE generated during 2050 (11.4 kt) will increase by more than 3 folds compared to 2022 estimates (3.3 kt). The average copper content in the WEEE stream fluctuates around the mean of 6.3 ± 0.3%. Most of the copper comes from the disposal of AC units with a percentage of 61% in 2022, increasing to 68.2% before decreasing to 66.6% and 63.6% in 2030, 2040, and 2050, respectively. The copper contents in the discarded Rs and Ws are steadily increasing with time from 14.2% and 6.5% in 2022 to approximately 17.6% and 12.3% in 2050, respectively. Yet, the share of discarded TVs in the WEEE complex is projected to contain less copper in the future with a percentage falling from 18.1% to 6.4% between 2022 and 2050. More details are illustrated in Figure 3.

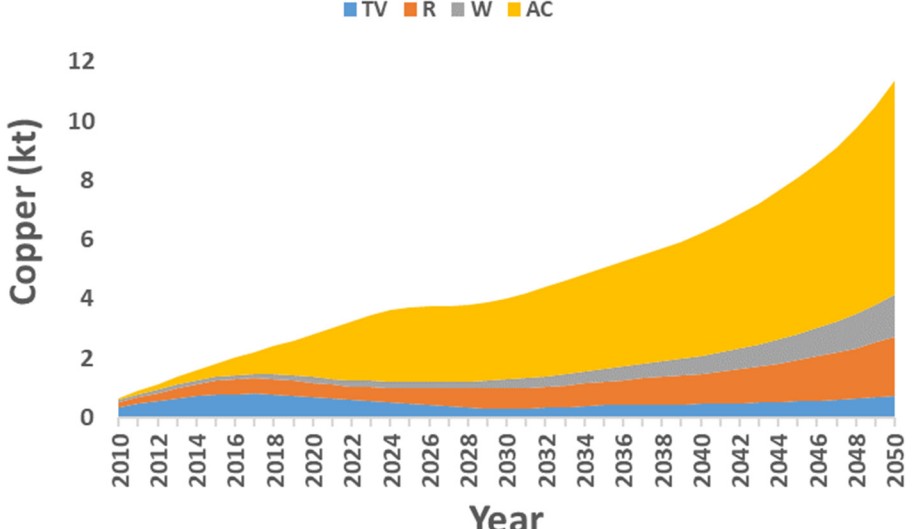

**Figure 3.** Copper contained in the discarded WEEE in Jordan.

Aluminum contained in WEEE forms approximately 4.8 ± 0.1% on average. It follows a similar trend to that of copper in WEEE. The details are illustrated in Figure 4. Most aluminum in the waste originates from ACs with a percentage of 52.2 ± 2.1% in the studied period. While the mean percentages in the other EEE are TV 13.6 ± 3.8%, R (15.6 ± 1)%, and W (18.5 ± 3.9)%.

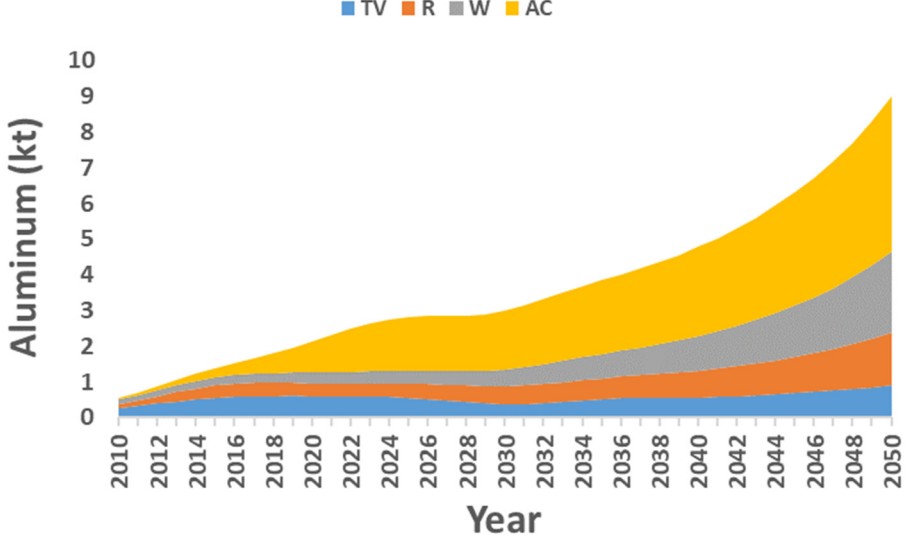

**Figure 4.** Aluminum contained in discarded WEEE in Jordan.

Precious metals such as silver and gold, on the other hand, are primarily present in low concentrations within the printed circuit boards (PCBs) of television sets and in the control circuits of other EEE. As shown in Figure 5, it is estimated that more than 600 kg and approximately 1130 kg of silver can be generated from the WEEE in 2022 and 2050, respectively. Similarly, gold content in WEEE will increase from 223 kg in 2022 to 475 kg in 2050.

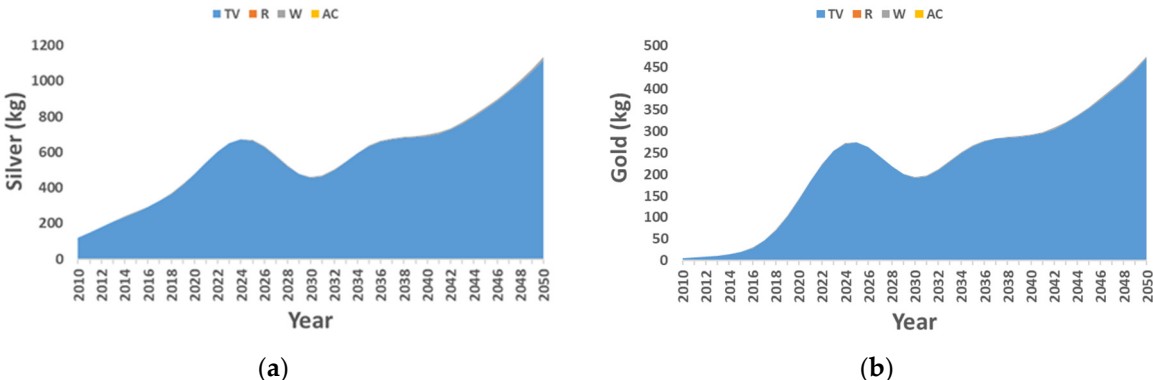

**Figure 5.** Silver (**a**) and gold (**b**) contained in the discarded WEEE in Jordan.

In contrast to precious metals, the percentage of plastics contained in WEEE is high. Its concentration comes after ferrous metals with an average 18 ± 0.8% by weight of WEEE. Plastics are used mostly in Rs, ACs, and TVs with a share percentage of more than 89% of the total plastics in WEEE. The projected quantities are shown in Figure 6. In refrigerators, plastic forms approximately 46.6 ± 3.6%, and it is used for door seals, drawers, lampshades, hinges, display panels, and shaft sleeves. More details about WEEE generation and material flow are shown in Table S1 in the Supplementary Material.

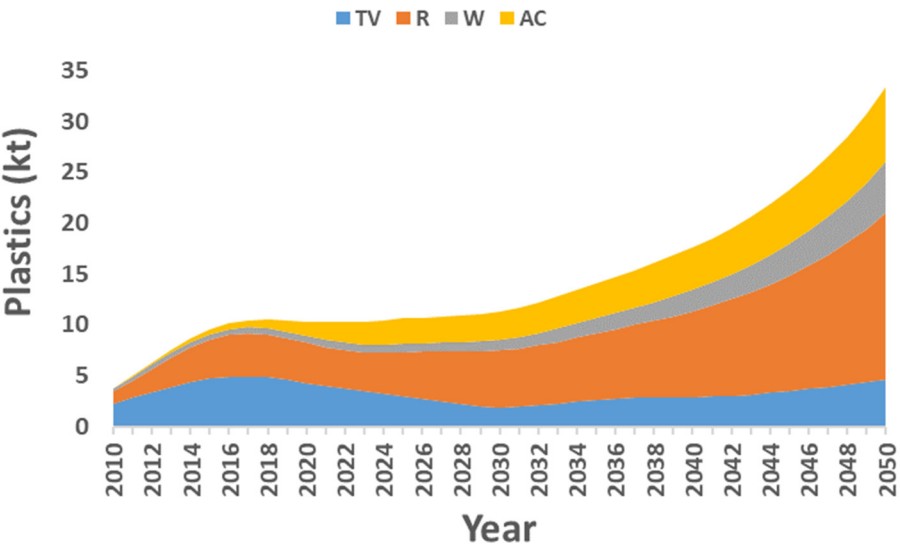

**Figure 6.** Plastics contained in the discarded WEEE in Jordan.

*3.3. Potential Number of WEEE Processing Plants*

The establishment of WEEE recycling plants to cope with the projected waste and material flow characteristics in the country is crucial. As explained in Section 2.5, the capacity of the WEEE treatment plant is approximately 0.5 million units per year in the present technology with a relative cost of approximately USD 25 million. Applying these conditions to Jordan WEEE generation projections, the estimated number of plants required, and its relative cost under baseline scenario can be approximated as shown in Table 5. By 2030, it is speculated that generated WEEE will require approximately four plants to treat this waste with a relative installation cost of approximately USD 100 million. Also, as highlighted earlier, the rate of waste generation is increasing, so the number of treatment facilities will increase accordingly to five and nine plants by 2040 and 2050, respectively.

**Table 5.** Number of WEEE processing plants required with relative installation cost.

| Year | Number of WEEE (Million Units) | Number of Required Facilities (Unit) | Cost (Million USD) |
|------|-------------------------------|--------------------------------------|--------------------|
| 2030 | 1.7 | 4 | 100 |
| 2040 | 2.6 | 5 | 125 |
| 2050 | 4.8 | 9 | 225 |

*3.4. Potential Financial Revenues of WEEE Recycling*

Given that the WEEE generation rates, with the corresponding flow of materials and the required treatment plants, are set, the question is what are the expected financial returns of this proposal? The relation of material value present in WEEE with time under baseline scenario is illustrated in Figure 7. This is based on the idealized case of recycling all materials contained in WEEE and average material market prices. However, taking the legal or actual recycling efficiencies are more practical. The financial revenues of recycling material contained in WEEE with time are recalculated and illustrated in Figure 8. They are calculated assuming legal efficiency ($\eta_L$) mandated in HARL, and material efficiency ($\eta_M$) obtained by PETEC.

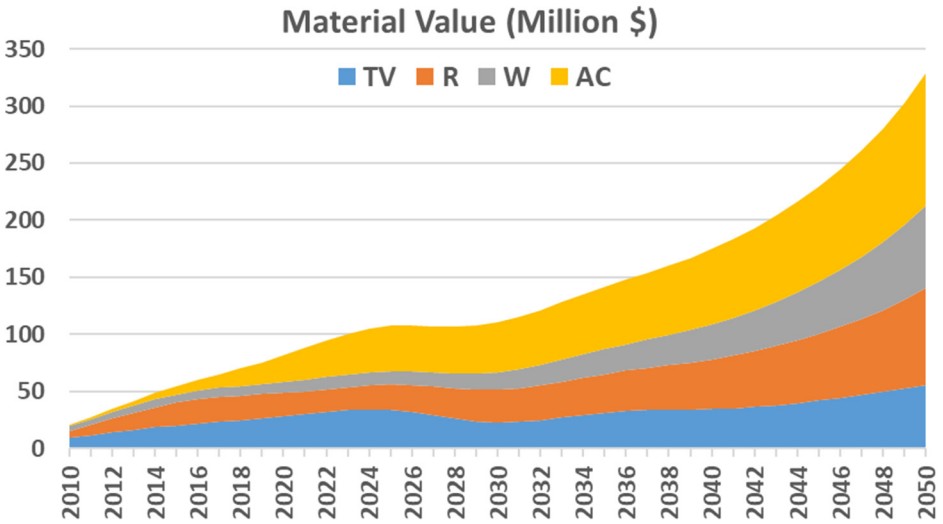

**Figure 7.** Variation in material value present in WEEE with time.

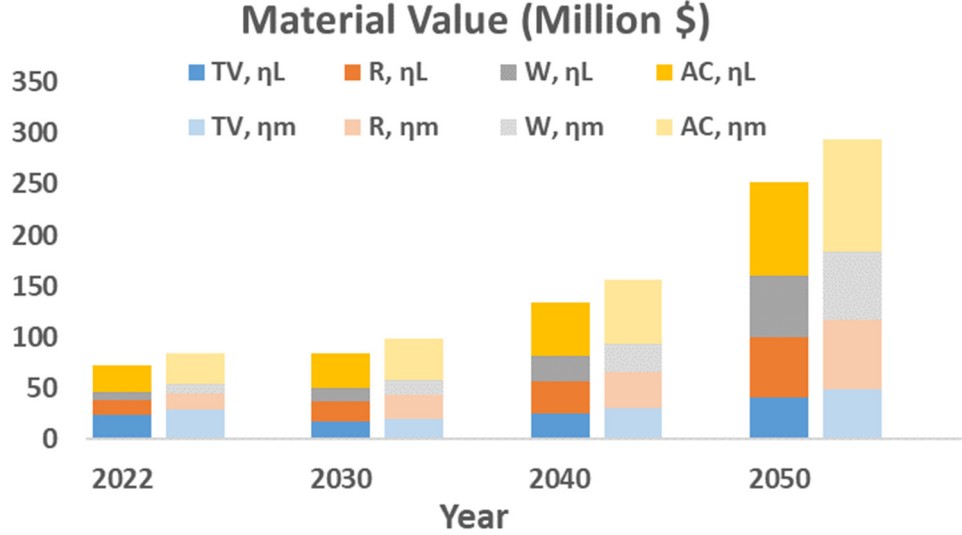

**Figure 8.** Potential material value contained in WEEE according to the legal ($\eta_L$) and material ($\eta_M$) efficiencies.

Projected revenues from the sale of recycled materials are growing over time, using 2022 as the base year for material market prices. The revenues of recycling plants that meet legal efficiency standards are projected to increase from over USD 72 million in 2022 to over USD 252 million in 2050, representing a revenue growth of 3.5 folds the 2022 estimates, whereas, if the facility complies to PETEC's recycling efficiency, it is projected to generate revenues of approximately USD 294 million in 2050, compared to the predicted USD 85 million in 2022. As anticipated, the PETEC process yields larger earnings compared to a system that just meets legal criteria, owing to its superior material recovery.

Furthermore, the income generated from recycling any product that meets the necessary legal criteria also serves as a reliable indicator of which equipment is more financially beneficial. In 2022, the potential revenues generated from recycling TVs and ACs surpassed 68%, with TVs accounting for 33% and ACs accounting for 35%. The remaining revenue shares were attributed to Rs at 19% and Ws at 13%. By 2050, the revenues generated from recycling the examined equipment will be distributed as follows: 16% for TVs, 23% for Rs, 24% for Ws, and 37% for ACs. Still ACs remain the most lucrative equipment for recycling, with profits above USD 92 million. The total income expected to be generated from the

treatment of Rs and Ws amounts to approximately USD 119 million. Recycling TVs can yield a potential revenue of over USD 41 million.

Another indicator to investigate is the revenues of WEEE per unit weight or per item. Both are useful to have insight into what is the financial value of waste and for determining the most efficient device for recycling. Table 6 summarizes the obtained results. The prices of obsolete appliances are arranged in descending order as follows R (USD 100 per item) > W (USD 72 per item) > AC (USD 70 per item) > TV (USD 31 for CRT TV, and USD 43 for FPD TV). On the other hand, the price per kg of each item can also be used for comparison. The prices of recyclable materials contained in FPD TV and AC units are USD 2.9 and USD 2.5 per kilogram, respectively, whereas the values for the other categories are USD 1.4, USD 1, and USD 0.9 per kilogram of WEEE generated from Rs, Ws, and CRT TVs, respectively. These indicators can be used as guidance for future valuation of discarded WEEE.

**Table 6.** Financial value and environmental impact of materials contained in obsolete EEE under baseline scenario.

| EEE Type | Price (USD/kg) | Price (USD/Item) | Environmental Impact (pt/kg) | Environmental Impact (pt/Item) |
|---|---|---|---|---|
| CRT TV | 0.95 | 31.42 | 0.19 | 6.34 |
| FPD TV | 2.90 | 42.63 | 0.31 | 4.59 |
| R | 1.44 | 99.73 | 0.29 | 20.36 |
| W | 0.99 | 71.86 | 0.18 | 13.01 |
| AC | 2.49 | 69.69 | 0.36 | 10.11 |

*3.5. Environmental Benefits of WEEE Recycling*

As mentioned before, in addition to the financial revenues of recycling WEEE, there is another benefit represented by avoiding the environmental impact of raw material production. Using the principles of life cycle assessment (LCA), the environmental impact indicator from the "ReCiPe" method for each element is calculated and summarized in Table 6. ACs have the greatest impact on the environment (0.36 pt/kg), followed by FPD TVs (0.31 pt/kg), Rs (0.29 pt/kg), CRT TVs (0.19 pt/kg), and finally Ws (0.18 pt/kg). So, for the same weight of waste, recycling ACs should be given priority to minimize the environmental impact resulting from its constituent's mining and production. The Environmental impact per unit item is arranged as follows: R > W > AC > CRT TV < FPD TV.

*3.6. Sensitivity Analysis*

3.6.1. Average EEE Weight and Composition Scenarios

The potential amount of WEEE generated from the baseline scenario (BL) and the two additional weight and composition scenarios (S1 and S2) are shown in Figure 9. The waste generated from S1 is higher than the BL scenario. The maximum relative difference is 14.3% of the BL scenario. On the contrary, the WEEE predicted from S2 is lower than the BL scenario. The minimum relative difference is 14.9% of the BL scenario. The mean relative difference between the predicted waste from the BL scenario and that from the two scenarios is within ±10% of the BL scenario.

On the other hand, the effect of varying composition scenarios influences the expected revenues from recycling WEEE over time. For example, scenarios S1 and S2 vary in the range of 75% to 120% and 58% to 109% of the BL scenario in the predicted period, respectively. The mean *RD* between the BL and the other scenarios is −7.5% and −28.5% for S1 and S2, respectively.

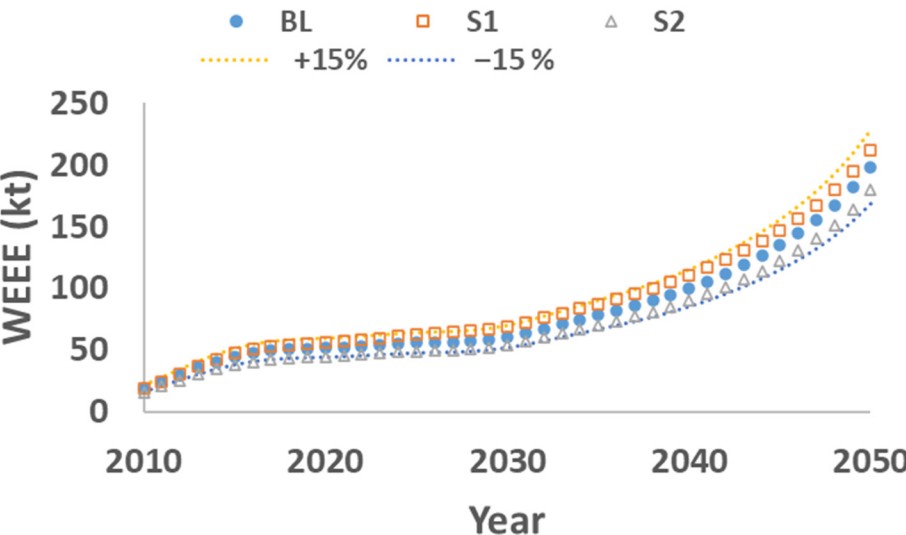

**Figure 9.** The predicted amount of WEEE generated using different scenarios and the ±15% of the baseline scenario shown in dotted lines.

### 3.6.2. Varying EEE Lifetime Parameters

The effect of varying EEE lifetime Weibull distribution parameters (the scale parameter, $\alpha$; and shape parameter, $\beta$) are investigated by taking ±30% of its original value in the BL scenario for each parameter. The forecast WEEE following the different scenarios is shown in Figure 10. It is clearly shown that the scale parameter greatly affects the generated WEEE. Increasing the lifetime of equipment, $\alpha$, by 30% reduces the average amount of generated WEEE by 20% and 18.8% by number and weight compared with the BL scenario, respectively. The range of *RD* of the EEE disposed of using this scenario is −50.3% to +0.5% by number (−50.4% to −2% by weight). And vice versa, the waste is increased when the scale parameter, $\alpha$, is reduced by 30%. On average, the *RD* of the number of EEE disposed of is 25.7% greater than the BL scenario. This corresponds to approximately 26% by weight. The maximum and minimum *RD* are 88.5% and −9.2% by number (93.5% and −7.2% by weight) compared with the BL scenario, respectively. This highlights the importance of this key parameter in estimating WEEE.

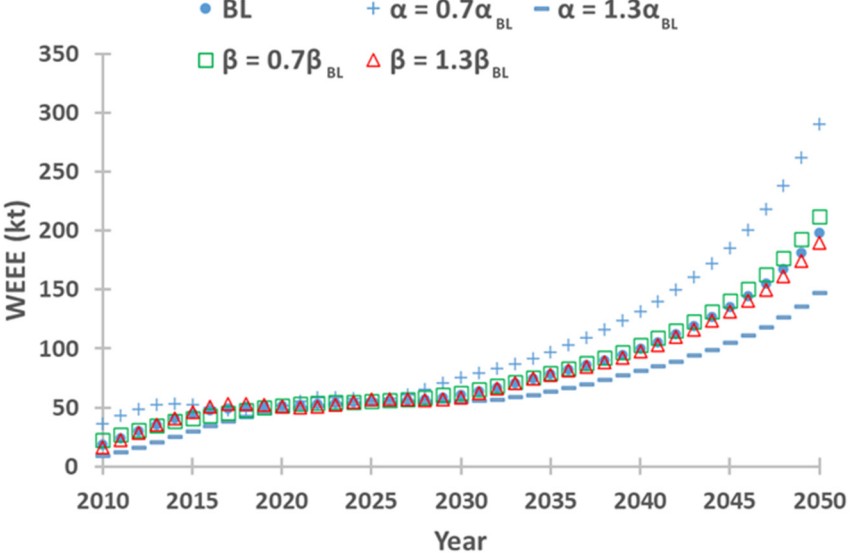

**Figure 10.** The effect of varying scale and shape parameters of the Weibull distribution function on the predicted amount of WEEE generation.

The effect of changing the shape factor, $\beta$, by $\pm30\%$ of the BL scenario has a relatively negligible influence on WEEE generation. By increasing the shape factor by 30%, the average amount of generated WEEE has been reduced by approximately 1% and 1.4% by number and weight compared with the BL scenario, respectively. Decreasing the shape factor by 30% has resulted in an average of 0.9% and 1.6% more waste by number and weight compared with the BL scenario, respectively.

The effect of changing the scale parameter affects the revenues as will be envisaged. On average the revenues rise by 29% when reducing the scale parameter by 30%. Revenues double BL scenario figures at certain periods, as well as reach 98% of BL scenario revenues. On the contrary, increasing the scale parameter by 30% causes a reduction in revenues compared with BL scenario revenues by approximately 21%, on average. Changing the shape parameter by 30% changes the revenues by approximately 1.7%.

## 4. Discussion

In this paper, the material flow from projected WEEE is estimated. Aligning with the global trend, the generation of WEEE in Jordan is on the rise. This is mostly attributed to increasing put on market appliances, changing lifestyle, and advancements in technology that have resulted in products becoming obsolete at a rapid pace. The rate of change in the material flow from the generated WEEE in Jordan varies with time due to two important factors. Firstly, the flood of approximately 1.4 million Syrian refugees fleeing their country due to the Syrian conflict started in 2011, which led to an increased demand for EEE in Jordan. Secondly, the spread of COVID-19 pandemic in 2019, caused a reduction in the EEE demand of the investigated appliances [21]. The PBM was able to capture the market dynamics in terms of delayed WEEE generation. The growing rate of WEEE in the country requires setting and executing appropriate strategies and policies to effectively manage this stream.

Generally, an increasing trend of material flow enclosed in the generated WEEE with time is observed. The average percentage of the materials are $54.5 \pm 1.4\%$, $6.3 \pm 0.3\%$, $4.8 \pm 0.1\%$, and $18 \pm 0.8\%$ for steel, copper, aluminum, and plastics, respectively. The precious metals (silver and gold) are found in small concentrations mainly in the PCB and in control circuits of other EEE. This indicates that the steel content is the highest share in the WEEE generated from household essential appliances, followed by plastics, copper, aluminum as observed by other researchers [32,58–60].

Jordan is considered one of the important countries in producing and exporting phosphate and potash minerals. In addition to construction and decorative stones, glass sand, and other nonmetallic minerals account for most of the mining output. Uranium, tin, copper, gold, and nickel all exist, as are several other metals of great economic worth. However, the absence of economic reserves of base metals, such as iron and aluminum, or not yet utilized reserves, such as copper and gold, due to immediate environmental and topographical constraints, it is crucial that the metals present in WEEE are consistently recovered.

Depletion of resources, vulnerability in supply chains, reliance on external metal sources, increasing metal prices, and elevated shipping costs are all drives in favor of recycling WEEE. The presence of ferrous, non-ferrous, and some critical metals in WEEE can be considered as an opportunity to convert this waste to a resource in line with the circular economy principles. Recycling and reusing WEEE can partially fulfill Jordan's need for these metals, minimize disruption often encountered in supply chains, reduce external debits toward importing raw materials, and improve the country's overall economic status, which is considered a prospective growth to the country.

Also, in addition to ensuring a local source of recyclable materials, WEEE is considered hazardous waste that should be managed. Hence, environmental mitigation is central in the form of neutralizing harmful substances from percolation to the environment and food chain, as well as reducing the amount of waste that requires landfilling. Furthermore, it aims to reduce the human-made earth disturbance that takes place during mining and mineral operations, which has negative impacts on ecosystems. These activities also result

in the release of significant amounts of greenhouse gas emissions and need a substantial amount of energy, typically derived from nonrenewable sources.

Jordan, through its reform agenda, has set policies, strategies, and plans to enhance reform matrix, economic priorities and modernizations, sustainable development plan, green economy, and environmental conservation. The Ministry of Environment (MoEnv) in Jordan recently approved instruction to manage WEEE [61]. It authorized a few companies to perform certain activities including WEEE collection and sorting. According to the WEEE instruction, the companies are obligated to report their activities to the MoEnv. Currently, due to lack of processing plants in Jordan, they export the collected WEEE to countries that have processing capacities under the supervision of the MoEnv. So, in order to plan ahead and provide the decision makers with suggestions and recommendations regarding future establishment of recycling operations, the approximated number of processing plants required were assessed by measuring the potential material flow from the WEEE generation estimates. The findings revealed that by the end of 2030, four recycling plants are required to handle the potential generated WEEE with approximated installation cost approximately USD 100 million based on Japanese example.

The expected revenues selling recycled materials are increasing with time. For recycling plants satisfying legal requirements, the revenues increase from approximately USD 72 million in 2022 to more than USD 252 million in 2050.

The environmental impact indicator (in pt/kg) showed that the order of investigated EEE is arranged in descending order as: ACs > FPD TV > R > CRT TV > W. So, for the same weight of waste, recycling ACs should be given priority.

In addition, the findings of the sensitivity analysis varying the EEE weight and composition revealed that the temporal variation in WEEE generation is within ±15%, and the mean relative difference between the predicted waste from the various scenarios is within ±10%. Also, it showed that the scale parameter is playing a fundamental role in the generated WEEE. The correlation between the scale parameter and the generated waste is inversely proportional. Increasing scale parameter by 30% reduces the generated waste by approximately 20% on average. And vice versa, decreasing scale parameter by 30%, increases the mean generation of WEEE by 25.7%. On the contrary, shape parameter has minimal effect.

In addition to the environmental and economic benefits, there are social advantages to reconsidering WEEE. Sustainable WEEE management needs a comprehensive system to deal with this stream from maintaining EEE inventory, collection, sorting, repair, refurbishment, reuse, dismantling, remanufacturing, recycling, incinerating, and disposal of useless/hazardous waste safely. Such multilevel, different specialties and duties, as well as diverse technical backgrounds certainly necessitate a trained workforce, establishing infrastructure to facilitate the missions of each party and ensuring technology transfer. For sure, these different horizons will represent an excellent opportunity for public-private partnerships (PPPs) to create jobs, fight unemployment, elevate the standard of living, as well as bypass the probabilistic illness that may occur from environmental pollution caused by improper management of WEEE. These cumulative visions of WEEE management will definitely improve Jordan's status in attaining different outcomes of the United Nations sustainable development goals (SDGs).

Ideally, it is recognized that the WEEE generation should include all EEE. However, in this study only the most used household appliances were considered. This somehow may be considered as a limitation of this study. Nevertheless, when it comes to reality, these EEE have the highest percentage share in society, and they are bulky and require space for storage once reach their EoL, hence, high collection rates are expected. All these reasons encourage the adoption of these EEE in this study.

## 5. Conclusions

WEEE recycling is gaining popularity as a means of enhancing resource efficiency, reducing waste environmental effect, and stimulating the circular economy. This study

employed PBM to estimate potential material flow from generated WEEE in Jordan for the first time. It approximated the required number of recycling plants to handle WEEE and showed the expected revenues based on applying the Japanese example as a whole package. The robustness and reliability of the findings were assessed using sensitivity analysis by examining how variations in appliances weight, composition, and lifetime parameters affect the generated waste. Though, there are some limitations to this study. For example, only four main household appliances (TVs, refrigerators, washing machines, and air conditioners) were included. The other limitation concerns the use of appliances average weight, composition, and lifetime. These limitations may affect the overall WEEE estimation. However, given the data-constrained environment in the country, the findings of this paper can be considered as a preliminary reference that may be used to guide proper planning for WEEE recycling and sustainable management.

**Supplementary Materials:** The following supporting information can be downloaded at: https://www.mdpi.com/article/10.3390/recycling9010004/s1, Table S1: Supplementary material for the baseline scenario.

**Author Contributions:** Conceptualization, L.A.A.-K. and F.Y.F.; Methodology, L.A.A.-K.; Validation, F.Y.F.; Formal analysis, L.A.A.-K. and F.Y.F.; Investigation, F.Y.F.; Resources, L.A.A.-K. and F.Y.F.; Writing — original draft, L.A.A.-K. and F.Y.F. Writing — review & editing, L.A.A.-K. and F.Y.F. All authors have read and agreed to the published version of the manuscript.

**Funding:** This research received no external funding.

**Data Availability Statement:** Data is contained within the Supplementary Materials. Further inquiries can be directed to the corresponding author.

**Conflicts of Interest:** The authors declare no conflict of interest.

## Appendix A

In this appendix, a summary of the data for the sensitivity analysis is presented. Tables A1 and A2 illustrate the average weight and composition of scenarios S1 and S2, respectively. The ±30% of the Weibull distribution function parameters of the EEE lifetime is detailed in Table A3.

**Table A1.** Average weight and composition of scenarios S1.

| EEE Type | Average Weight (kg) | Fe (%) | Stainless Steel | Cu (%) | Al (%) | Ag (PPM) | Au (PPM) | Pd (PPM) | Plastics (%) | Others (%) | Zn (%) | In (PPM) |
|---|---|---|---|---|---|---|---|---|---|---|---|---|
| CRT TV | 33.2 | 10.3 | NR | 3.7 | 2.6 | 12 | 0.5 | 2 | 22.8 | 60.6 | NR | NR |
| LCD TV | 9.5 | 28.1 | 1.34 | 1.5 | 4.79 | NR | 2.42 | NR | NR | NR | 0.43 | 530 |
| R | 85.8 | 45.1 | 1.14 | 3.39 | 1.22 | NR | 0.21 | NR | NR | NR | 0.38 | NR |
| W | 48.1 | 40.6 | 20.8 | 29 | 0.2 | NR | 0.52 | NR | NR | NR | 0.48 | NR |
| AC | 47.1 | 45.7 | 0.22 | 1.49 | 7.71 | NR | 0.68 | NR | NR | NR | 0.48 | NR |

**Table A2.** Average weight and composition of scenarios S2.

| EEE Type | Average Weight (kg) | Fe (%) | Stainless Steel | Cu (%) | Al (%) | Ag (PPM) | Au (PPM) | Plastics (%) | Zinc (%) |
|---|---|---|---|---|---|---|---|---|---|
| CRT TV | 28.3 | 22.8 | NR | 4.3 | 3.9 | 48.1 | 8.0 | 15.0 | 0.1 |
| LCD TV | 13.9 | 22.8 | NR | 4.3 | 3.9 | 48.1 | 8.0 | 15.0 | 0.1 |
| R | 50.6 | 63.1 | 2.6 | 2.5 | 3.2 | NR | NR | 8.5 | NR |
| W | 72.2 | 54.2 | 1.7 | 3.2 | 1.7 | NR | NR | 15.7 | NR |
| AC | 30.9 | 63.1 | 2.6 | 2.5 | 3.2 | NR | NR | 8.5 | NR |

**Table A3.** The ±30% of the Weibull distribution function parameters of the EEE lifetime.

| EEE Type | Baseline | | $\alpha$ | | | | $\beta$ | | | |
|---|---|---|---|---|---|---|---|---|---|---|
| | | | +30% $\alpha_{BL}$ | | −30% $\alpha_{BL}$ | | +30% $\beta_{BL}$ | | −30% $\beta_{BL}$ | |
| | $\alpha$ | $\beta$ | $\alpha$ | $\beta$ | $\alpha$ | $\beta$ | $\alpha$ | $\beta$ | $\alpha$ | $\beta$ |
| TV | 10.5 | 3.44 | 13.65 | 3.44 | 9.6 | 3.44 | 10.5 | 4.47 | 10.5 | 2.41 |
| R | 11.8 | 3.44 | 15.34 | 3.44 | 10.7 | 3.44 | 11.8 | 4.47 | 11.8 | 2.41 |
| W | 9.3 | 3.44 | 12.09 | 3.44 | 8.5 | 3.44 | 9.3 | 4.47 | 9.3 | 2.41 |
| AC | 8.3 | 3.44 | 10.79 | 3.44 | 7.6 | 3.44 | 8.3 | 4.47 | 8.3 | 2.41 |

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
