# Peer review of "The Potential Material Flow of WEEE in a Data-Constrained Environment—The Case of Jordan"

_recycling, doi:10.3390/recycling9010004_

Round 1
Reviewer 1 Report
Comments and Suggestions for Authors
Paper :
The potential material flow of WEEE in a data-constrained evironment. The case of Jordan
In this paper, a model based on population balance is utilized to assess the potential material flow characteristics from the WEEE in Jordan.
The authors' goal was:
1.quantify the material flow expected from recycling the generated WEEE,
2. proposes the number of plants required to recycle this secondary resource,
3. and outlines the expected economic and environmental benefits that could be achieved from recycling operations.
The question is, was it successful? In my opinion, only the first goal was achieved. There is a long way from determining the flow of waste (WEEE) to its recycling, especially in a country without a tradition of using pyro and hydrometallurgy.
However, Sensitivity Analysis and Environmental Impact should be appreciated.
​
1. come from the 13 rapid increase in demand for appliances and the decreasing lifetimes of gadgets.
WEEE is not just “gadgets”, it is equipment, devices not only for fun, it is a scientific article, not an advertising paper.
2. The expected metal content in WEEE in 2022 is around 26 kt, 3.3 kt, and 2.5 kt, respectively. These 21
Where did this data come from? provide the sources (if they are in the introduction then Ok)
3. ……..ization, industry 4, and technological advancement lead to rapid product obsolescence 33
industry 4.0
4. Base metals, including iron, steel, aluminum, copper, 47
zinc, and their alloys, form the majority by weight. Precious metals, such as gold, silver, 48and palladium, are used in low proportions especially in printed circuit boards (PCBs), 49while plastics represent (15 – 35) percent of WEEE [6]–[8].
It is worth mentioning critical metals (e.g. REM, etc.), without which today's EEE cannot work at the appropriate level of advancement.
5. In this paper, the main objective is to assess the potential material flow of recycling 90 WEEE in Jordan.
The article is of local importance because the problems it raises are already known on a global scale. But it's good that developing countries are thinking about protecting the Earth.
6. These include, but 99
are not limited to, the absence of reliable waste inventory records, inefficient or uncom-
100 prehensive regulations controlling waste handling/treatment/disposal, lack of
appropri- 101 ate technological infrastructure, and deficiency of financial support [15],
[16].
It is good that the article will determine the amount and flow of WEEE, and recycling can be carried out in countries with advanced processing capabilities - avoiding wild recycling.
7. In this paper, PBM is used to assess the flow of EEE quantities in Jordan. Based on 117
diffusion rates obtained using the logistic growth model [16] and penetration rates in
Jor- 118danian society [31],
consumption volume?
8. High rates of diffusion of these 133
gadgets have been observed in developing countries during economic growth periods 134 equpments
of these 133 gadgets have been observed in developing countries during economic growth periods 134
it is slang, a refrigerator or an air conditioner are not gadgets
9. It is success- 146
fully applied to estimate WEEE in developing countries such as South Korea [28],
Vietnam 147 Recycling 2023, 8, x FOR PEER REVIEW 4 of 17[27] and Jordan [16].
South Korea is a developed country – member og G20!
10. Indeed, Jordan can benefit from the accumulative expertise of developed countries 184
……..
recommended for building any future processing plants in Jordan, hence it is adopted in 195
this study. 196
Is this related to the content of the subsection title? 2.4. Materials included in WEEE
11. Please check whether the content agrees with the titles of chapters and sub-chapters, you cannot write about everything, it should be organized and multiple repetitions should be eliminated.
Author Response
We want to thank the reviewer for her/his positive and insightful comments on the manuscript. Below is our response to the issues raised in the review in the attached file.

Reviewer 2 Report
Comments and Suggestions for Authors
Congratulation for the paper, the topic is in line with our days.
About the paper structure it needs to be more clear. You started with a prediction of WEEE generation until 2050, then you suddenly entered the impact of this, two scenarios appeared that were not previously described.
Likewise, any methodology/method used must be described previously, or with reference to it.
From my point of view you need to improve the structure and to follow your objectives. In this form the manuscript can no be published.

Author Response

(The authors gave the same response as above.)

Reviewer 3 Report
Comments and Suggestions for Authors
The work is interesting to make prospective in relation to the recycling potential of waste byproducts of electrical and electronic origin. WEEE
The study is correctly justified in the introductory part and the bibliographic review justifying the method is sufficient. The results are well presented.
However, it would be necessary to make some justifications before it could be published.
It would be necessary to explain more clearly the PBC method used. It should be justified with more studies or scientific works published in a high-impact journal Q1 or Q2 to give scientific solvency to the method.
It would also be necessary to clarify the reason for choosing the types of products selected to apply the method. Why or under what criteria have you used or selected the references (CRT TV, FPD TV. R., AC)? This needs to be clearly justified.
The results are shown in figures but the supporting tables are not presented. They should be provided at least as complementary material.
On the other hand, it would be more interesting to reconsider separating the Results and Discussion sections in the document more clearly.
The Conclusions should be rewritten, relate them to the objectives set and clearly indicate the novelty or scientific contribution of the study, as well as its impact and orientation for future lines of research in a more clear and concrete way.
It is recommended to review the bibliography and adapt it to the format of the magazine. For example:
[32] oracle, “Department of Statistics Jordan | Home,” العامة Ø§Ù„Ø¥ØØµØ§Ø¡Ø§Øª دائرة. Accessed: Nov. 22, 2023. [Online]. Available: 588
https://dosweb.dos.gov.jo/
Comments on the Quality of English LanguageThe work is interesting to make prospective in relation to the recycling potential of waste byproducts of electrical and electronic origin. WEEE
The study is correctly justified in the introductory part and the bibliographic review justifying the method is sufficient. The results are well presented.
However, it would be necessary to make some justifications before it could be published.
It would be necessary to explain more clearly the PBC method used. It should be justified with more studies or scientific works published in a high-impact journal Q1 or Q2 to give scientific solvency to the method.
It would also be necessary to clarify the reason for choosing the types of products selected to apply the method. Why or under what criteria have you used or selected the references (CRT TV, FPD TV. R., AC)? This needs to be clearly justified.
The results are shown in figures but the supporting tables are not presented. They should be provided at least as complementary material.
On the other hand, it would be more interesting to reconsider separating the Results and Discussion sections in the document more clearly.
The Conclusions should be rewritten, relate them to the objectives set and clearly indicate the novelty or scientific contribution of the study, as well as its impact and orientation for future lines of research in a more clear and concrete way.
It is recommended to review the bibliography and adapt it to the format of the magazine. For example:
[32] oracle, “Department of Statistics Jordan | Home,” العامة Ø§Ù„Ø¥ØØµØ§Ø¡Ø§Øª دائرة. Accessed: Nov. 22, 2023. [Online]. Available: 588
https://dosweb.dos.gov.jo/
Author Response

(The authors gave the same response as above.)

Round 2
Reviewer 2 Report
Comments and Suggestions for Authors
Thank you for the revised paper.
Is a good paper, you presented to much information and data that does not always make sense and continuity.
Regarding the LCA methodology you have to define the four steps and to cite not only the method you used but also the methodology https://www.sciencedirect.com/topics/engineering/life-cycle-assessment-methodology.
Author Response
Thank you for your valuable comments. We appreciate your time and efforts.

Reviewer 3 Report
Comments and Suggestions for Authors
The authors satisfactorily answered
Comments on the Quality of English LanguageThe author responded satisfatoryly
Author Response

(The authors gave the same response as above.)
